# Fetal biparietal diameter as a potential risk factor for prolonged second stage of labor: A retrospective observational cohort study

Satoshi Shinohara[1]*, Atsuhito Amemiya[1], Motoi Takizawa[1], Kohta Suzuki[2]

1 Department of Obstetrics and Gynecology, National Hospital Organization Kofu National Hospital, Kofu, Yamanashi, Japan, 2 Department of Health and Psychosocial Medicine, Aichi Medical University School of Medicine, Nagakute, Aichi, Japan

* shinohara617@gmail.com

**Data Availability Statement:** The data are all contained within the manuscript.

**Funding:** The authors received no specific funding for this work.

## Abstract

Prolonged second stage of labor is a common abnormality of labor progression. Very little research exists regarding the relationship between prolonged second stage of labor and antepartum sonographic fetal head biometry parameters, especially fetal biparietal diameter (BPD). Fetal BPD assessment is essential for estimating fetal weight, and these measurements are readily available to Japanese obstetricians. We conducted a retrospective observational cohort study to evaluate the association between BPD fetal Z-score and prolonged second stage of labor in a Japanese cohort. Individual BPD data measured using a GE Voluson 730 expert ultrasound system (GE, Healthcare Japan, Tokyo, Japan) were converted to Z-scores for a particular gestational age. After excluding patients with multiple pregnancies and emergency or elective cesarean sections, a total of 2,711 (nulliparity, n = 1341) Japanese women who delivered at term were included. We analyzed the incidence of prolonged second stage of labor and the association between BPD Z-score measured <7 days before delivery and prolonged second stage of labor by parity. The overall incidence of prolonged second stage of labor was 18.3% (246/1,341) in nulliparous women and 4.6% (63/1,370) in multiparous women. In nulliparous women, multivariable analysis indicated that BPD Z-score was significantly associated with prolonged second stage of labor (adjusted odds ratio, 1.18; 95% confidence interval, 1.02–1.37). Kaplan-Meier survival analysis showed that at each time point during the second stage of labor, the percentage of women who had not yet delivered was higher among those who delivered neonates with large BPD Z-scores than among those who delivered neonates with smaller BPD Z-scores. On the contrary, in multiparous women, BPD Z-score was not statistically associated with prolonged second stage of labor. Our results suggest that considering BPD Z-score is helpful in the management of nulliparous women who are at risk of developing a prolonged second stage of labor.

**Competing interests:** The authors have declared that no competing interests exist.

## Introduction

The second stage of labor is defined as the duration of time from full cervical dilatation to delivery of the new-born [1]. Prolonged second stage of labor is usually defined as 3 h for primiparous women with an epidural and 2 h in those without and as more than 2 h for parous women with an epidural and more than 1 h in those without [1,2]. Prolonged second stage of labor, which is associated with increased rates of maternal and fetal morbidities, is one of the most common types of "abnormality of labor progression." For the mother, prolonged second stage of labor increases the risk of operative vaginal delivery using either forceps or vacuum device, third-/fourth-degree perineal lacerations, postpartum hemorrhage, postpartum urinary retention, and spontaneous delivery earlier than 37 weeks of gestation in the subsequent pregnancy [3–7]. Similarly, for the neonate, prolonged second stage of labor increases the risk of admission to the neonatal intensive care unit, neonatal seizures, and sepsis [8,9]. However, predicting the risk of prolonged second stage of labor in the clinical setting is difficult. Therefore, a complete understanding of the risk factors associated with prolonged second stage of labor and the development of appropriate treatment strategies are essential. Prolonged second stage of labor may be attributed to several factors, such as nulliparity, increasing maternal weight gain, use of regional anesthesia, induction of labor, fetal occiput in a posterior or transverse position, and increased birthweight [1,10–13]. Head circumference (HC), which is an antepartum sonographic fetal head biometry parameter, has been used to assess the risk of prolonged second stage of labor in recent years. A previous study has reported that large HC increased the risk of prolonged second stage of labor [1]. However, little research has been performed concerning the relationship between biparietal diameter (BPD) and prolonged second stage of labor [14]. Prenatal check-ups in Japan are conducted once per week from 36 to 39 weeks and twice per week after 40 weeks [15]. Although Japanese obstetricians do not necessarily calculate the HC, they do calculate the BPD at each prenatal check-up to obtain the estimated fetal weight using the Shinozuka technique [16]. The formula is as follows: EFW (g) = $1.07 \times BPD^3 + 3.00 \times 10^{-1}AC^2 \times FL$, where BPD stands for biparietal diameter (cm), AC stands for abdominal circumference (cm), and FL stands for femur length (cm) [16]. Consequently, in term pregnancies, sonographic measurements of BPD within 7 days before delivery are readily available to obstetricians in the clinical setting. Therefore, even with the same antepartum sonographic fetal head biometry parameter, it may be easier to use BPD to assess the risk for prolonged second stage of labor than to use HC in Japan. Improved understanding of the relationship between BPD before delivery and prolonged second stage of labor will provide clinically useful information for perinatal management in Japan. This is because, especially in Japan, there has been little research done concerning second stage of labor. Moreover, the risk factors for prolonged second stage of labor and evidence about the guide for the second stage of labor are insufficient [17].

Therefore, we performed a retrospective study in a Japanese cohort to clarify the potential effects of fetal BPD on the second stage of labor. The primary objective of this study was to investigate whether large BPD can be a risk factor for prolonged second stage of labor. Our secondary objective was to analyze whether BPD affects the duration of second stage of labor by survival analysis.

## Materials and methods

### Study design

We conducted a retrospective observational cohort study at the National Hospital Organization Kofu National Hospital between January 2012 and December 2019. Since the National Hospital

Organization Kofu National Hospital is one of the tertiary perinatal care centers in Yamanashi Prefecture, relatively high-risk patients were included in this study. The study included women with singleton pregnancies delivered at 37 + 0 to 41 + 6 weeks of gestation, whose sonographic measurements of BPD within 7 days before delivery were available. Women with multiple pregnancies (n = 24), those who underwent an elective or emergency cesarean section (n = 572), and those with missing data (n = 16) were excluded. Indications for operative vaginal delivery, which was performed to shorten and reduce the effects of the second stage of labor, included non-reassuring fetal status and lack of continuing labor progress for at least 2 h, regardless of a nulliparous or multiparous status. The study protocol was reviewed and approved by the Human Subjects Review Committee of the National Hospital Organization Kofu National Hospital, and the requirement for acquisition of informed consent from patients was waived owing to the retrospective study design. Nevertheless, patients were provided with the opportunity to refuse the use of their data through the hospital's website. All procedures were performed in accordance with the 1964 Helsinki Declaration and its later amendments.

## Data collection

We collected obstetric data from medical records. Gestational age was determined based on the maternally reported last menstrual period and was confirmed by the crown-rump length measured on the first-trimester sonogram. We recorded data on the mother's age at delivery, fetal sex, presence of gestational diabetes mellitus (GDM), presence of hypertensive disorder of pregnancy (HDP), use of *in vitro* fertilization (IVF), parity, gestational age at delivery, maternal stature, and pre-pregnancy weight status. Additionally, we assessed maternal weight gain, induction of labor, and macrosomia, which are potential confounding factors that have been reported to be risk factors for prolonged second stage of labor [1,10–13]. The dose and type of uterine contraction agent (oxytocin or prostaglandin) administered were determined by the treating obstetrician during labor induction or augmentation, according to the guidelines for obstetrical practice in Japan [15]. Our facility does not perform delivery with an epidural anesthesia. The fetal BPD was measured from the outer edge of the proximal calvaria to the inner edge of the distal calvarial wall (outer–inner) at the level of the third ventricle and thalami [14,18]. Prolonged second stage of labor was defined as labor exceeding 1 h after complete cervical dilatation in a multiparous woman and labor exceeding 2 h in a nulliparous woman [1]. A diagnosis of GDM was made if there was at least one abnormal plasma glucose value (≥92, 180, and 153 mg/dL as fasting, 1 h, and 2 h plasma glucose concentrations, respectively) after a 75 g oral glucose tolerance test [15]. HDP was defined as a blood pressure level ≥140/90 mmHg on at least two occasions [19]. Body mass index (BMI) before pregnancy was calculated based on World Health Organization standards, and patients were classified as obese (≥25.0 kg/m$^2$) or non-obese (<25.0 kg/m$^2$) according to the Japan Society of Obstetrics and Gynecology Guidelines for Obstetrical Practice 2014 [15]. Maternal weight gain during pregnancy was calculated by subtracting the patient's pre-pregnancy body weight from her body weight at the last prenatal visit before delivery. Excessive maternal weight gain was defined according to pre-pregnancy BMI. If the pre-pregnancy BMI was <25.0 kg/m$^2$, excessive weight gain was defined as weight gain of 12 kg or more throughout the pregnancy. If the pre-pregnancy BMI was ≥25.0 kg/m$^2$, excessive weight gain was defined as weight gain of 7 kg or more throughout the pregnancy [15]. Macrosomia was defined as a birth weight ≥3,500 g [20].

## Statistical analyses

First, individual BPD data of the study populations were converted to Z-score for a particular gestational age on the following formula; Z-score = (individual BPD–cohort mean) / (cohort

**Table 1. Reference of BPD value based on Z-score (-1.645 = 5th and 1.645 = 95th) every gestational week.**

| Gestational age, weeks | BPD value at each Z-score, mm | |
|---|---|---|
| | -1.645 (5%tile) | 1.645 (95%tile) |
| 37 weeks | 83.8 | 95.8 |
| 38 weeks | 85.1 | 96.7 |
| 39 weeks | 86.9 | 97.7 |
| 40 weeks | 86.5 | 98.7 |
| 41 weeks | 87.8 | 99.0 |

BPD, biparietal diameter.

standard deviation). This is because we thought that even with the same BPD value, the effect of BPD on labor process is different per gestational week. We described the reference of BPD value from Z-score (-1.645 = 5th and 1.645 = 95th) every gestational week to better understand the study results (Table 1).

Then, the Mann-Whitney U and Chi-squared tests were used to evaluate the effect of potential confounding factors for prolonged second stage of labor. Next, a multivariable logistic regression model was used to identify variables significantly associated with prolonged second stage of labor, because the purpose of this study was to investigate whether a clinically significant duration of second stage of labor is associated with a large BPD. Finally, we used Kaplan-Meier analysis and the log-rank test for examining the statistical differences based on the different Z-score of BPD in the duration of second stage of labor, starting from full cervical dilation.

When using the Kaplan-Meier analysis, we divided cases into three groups according to Z-score of BPD (or percentile) (below -1.645 (= 5th percentile), -1.645 to 1.645 (= 5th to 95th percentile), and above 1.645 (= 95th percentile)). All analyses were performed using Bell Curve for Excel (Social Survey Research Information Co., Ltd., Tokyo, Japan), and the significance level was set at $P<0.05$.

## Results

A total of 2,711 women were considered eligible for inclusion in this study. The mean maternal age was $31.1 \pm 5.3$ years, and the mean maternal pre-pregnancy BMI was $20.9 \pm 3.2 \text{ kg/m}^2$, with 1,341 nulliparous women (49.5%), 1,320 male infants (48.7%), 166 women with GDM (6.1%), 78 women with HDP (2.9%), and 29 macrosomic infants (1.1%). Table 2 summarizes the clinical characteristics of women enrolled in this study. The overall incidence rate of prolonged second stage of labor was 11.4% (309/2,711). The mean duration of second stage of labor was $45.8 \pm 56.7$ min. Nulliparity, gestational age, IVF, pre-treatment with uterine contraction agent, male sex, birth weight, instrumental delivery, fourth-degree perineal lacerations, and duration of second stage of labor were significantly higher in the prolonged second stage of labor group than in the normal second stage of labor group, whereas maternal age, maternal height, Apgar score (5 min), and UA (umbilical arterial) pH were significantly lower (Table 2).

### Relation between Z-score of BPD and prolonged second stage of labor in nulliparity

Table 3 summarizes the clinical characteristics of 1,341 nulliparous women enrolled in this study. The overall incidence of prolonged second stage of labor was 18.3% (246/1,341) in

**Table 2. Baseline characteristics of the study population.**

| Characteristic | | | P-value |
|---|---|---|---|
| Gestational age, weeks | 39.3 ± 1.05 | 39.6 ± 0.97 | <0.001 |
| Maternal height, cm | 158.4 ± 5.4 | 157.7 ± 5.3 | 0.02 |
| Pre-pregnancy BMI, kg/m² | 20.9 ± 3.2 | 20.5 ± 2.7 | 0.33 |
| IVF | 71 (3.0) | 22 (7.1) | <0.001 |
| Birth weight, g | 3,065 ± 361.9 | 3,170 ± 348.7 | <0.001 |
| HDP | 66 (2.7) | 12 (3.8) | 0.26 |
| GDM | 146 (6.1) | 20 (6.5) | 0.79 |
| Instrumental delivery | 49 (2.0) | 46 (14.8) | <0.001 |
| Pre-treatment with a uterine contraction agent | 656 (27.3) | 199 (64.4) | <0.001 |
| Male infant | 1159 (48.3) | 161 (52.1) | 0.20 |
| Duration of second stage of labor, min | 30.0 ± 26.7 | 169.1 ± 74.5 | <0.001 |
| Fourth-degree perineal lacerations | 2 (0.083) | 2 (6.5) | <0.001 |
| Apgar score, 5 min | 9.33 ± 0.65 | 9.22 ± 0.67 | 0.004 |
| UA pH | 7.32 ± 0.15 | 7.30 ± 0.07 | <0.001 |

Values are presented as average ± standard deviation or as numbers (%).

BMI, body mass index; IVF, in vitro fertilization; HDP, hypertensive disorder of pregnancy; GDM, gestational diabetes mellitus; UA, umbilical arterial.

nulliparity. Maternal age, gestational age, IVF, pre-treatment with uterine contraction agent, birth weight, instrumental delivery, fourth-degree perineal lacerations, and duration of second stage of labor were significantly higher in the prolonged second stage of labor group than in the normal second stage of labor group, whereas maternal height, Apgar score (5 min), and UA pH were significantly lower (Table 3).

There was a statistically significant difference in the rate of prolonged second stage of labor among the quartiles (Table 4). As demonstrated in Table 4, there were significant differences

**Table 3. Baseline characteristics of nulliparous women in the study population.**

| | Normal second stage of labor (n = 1,095) | Prolonged second stage of labor (n = 246) | P-value |
|---|---|---|---|
| Maternal age, years | 29.4 ± 5.4 | 31.5 ± 4.8 | < 0.001 |
| Gestational age, weeks | 39.4 ± 1.07 | 39.6 ± 0.99 | 0.006 |
| Maternal height, cm | 158.5 ± 5.4 | 157.5 ± 5.3 | 0.014 |
| Pre-pregnancy BMI, kg/m² | 20.5 ± 2.8 | 20.4 ± 2.6 | 0.27 |
| IVF | 40 (3.7) | 17 (6.9) | 0.02 |
| Birth weight, g | 3008 ± 342.6 | 3135 ± 332.1 | < 0.001 |
| HDP | 34 (3.1) | 10 (4.1) | 0.45 |
| GDM | 58 (5.3) | 18 (7.3) | 0.22 |
| Instrumental delivery | 44 (4.0) | 44 (17.8) | < 0.001 |
| Pre-treatment with a uterine contraction agent | 403 (36.8) | 173 (70.3) | < 0.001 |
| Male infant | 539 (49.2) | 123 (50.0) | 0.83 |
| Duration of second stage of labor, min | 48.4 ± 28.2 | 185.7 ± 65.1 | < 0.001 |
| Fourth-degree perineal lacerations | 2 (0.18) | 2 (0.81) | <0.001 |
| Apgar score, 5 min | 9.28 ± 0.70 | 9.21 ± 0.68 | 0.03 |
| UA pH | 7.30 ± 0.21 | 7.29 ± 0.07 | 0.006 |

Values are presented as average ± standard deviation or as numbers (%).

BMI, body mass index; IVF, in vitro fertilization; HDP, hypertensive disorder of pregnancy; GDM, gestational diabetes mellitus; UA, umbilical arterial.

**Table 4. Prevalence of prolonged second stage of labor according to Z-score of fetal biparietal diameter quartiles measured within 7 days before delivery in nulliparous women.**

|  | Group 1 (Z-score of BPD; -5.58~-0.54) n = 327 | Group 2 (Z-score of BPD; -0.53~0.04) n = 307 | Group 3 (Z-score of BPD; 0.05~0.76) n = 389 | Group 4 (Z-score of BPD; 0.77~3.11) n = 318 | P-value |
|---|---|---|---|---|---|
| Prolonged second stage of labor | 45 (13.8) | 52 (16.9) | 82 (21.1) | 67 (21.1) | 0.04 |

Values are presented as numbers (%). BPD, biparietal diameter.

in the rate of prolonged second stage of labor among the four groups of women, and the higher the Z-score of BPD value, the higher the rate of prolonged second stage of labor.

In multivariable analyses, the adjusted odds ratio (aOR) for prolonged second stage of labor was 1.18 (95% confidence interval [CI], 1.02 to 1.37; P<0.001) for each 1 increase in Z-score of BPD. Moreover, macrosomia (aOR, 1.63; 95% CI, 1.06–2.50) was found to be associated with prolonged second stage of labor (Table 5).

Using the Kaplan-Meier survival analysis (Fig 1), at each time point during the second stage of labor, the percentage of women who had not yet delivered was larger for women who delivered neonates with large Z-scores of BPD than those who delivered neonates with smaller Z-scores of BPD (median duration of second stage of labor 65.0 min vs 71.9 min vs 108.5 min, P < 0.001).

## Relation between Z-score of BPD and prolonged second stage of labor in multiparity

Table 6 summarizes the clinical characteristics of 1,370 multiparous women enrolled in this study. The overall incidence of prolonged second stage of labor was 4.6% (63/1,370) in multiparity. Maternal age, gestational age, IVF, pre-treatment with uterine contraction agent, birth weight, instrumental delivery, male infant, and duration of second stage of labor were significantly higher in the prolonged second stage of labor group than in the normal second stage of labor group, whereas UA pH was significantly lower (Table 6).

There was no statistically significant difference in the rate of prolonged second stage of labor among the quartiles (Table 7).

**Table 5. Factors associated with prolonged second stage of labor in nulliparous women.**

| Variables | Prolonged second stage of labor (n = 246) | Normal second stage of labor (n = 1,095) | Crude | | Adjusted | |
|---|---|---|---|---|---|---|
|  |  |  | OR | 95% CI | OR | 95% CI |
| Z-score of BPD (median [25th–75th percentile]) | 0.17 (-0.43~0.88) | 0.04 (-0.70~0.65) |  |  | 1.18 | 1.02–1.37 |
| Induction of labor |  |  |  |  |  |  |
| No | 193 | 912 | 1.0 | Reference | 1.0 | Reference |
| Yes | 53 | 183 | 1.37 | 0.97–1.93 | 1.34 | 0.95–1.90 |
| Excessive maternal weight gain |  |  |  |  |  |  |
| No | 106 | 432 | 1.0 | Reference | 1.0 | Reference |
| Yes | 140 | 663 | 1.16 | 0.88–1.54 | 1.10 | 0.83–1.46 |
| Macrosomia |  |  |  |  |  |  |
| No | 209 | 1003 | 1.0 | Reference | 1.0 | Reference |
| Yes | 37 | 92 | 1.93 | 1.28–2.91 | 1.63 | 1.06–2.50 |

BPD, biparietal diameter.

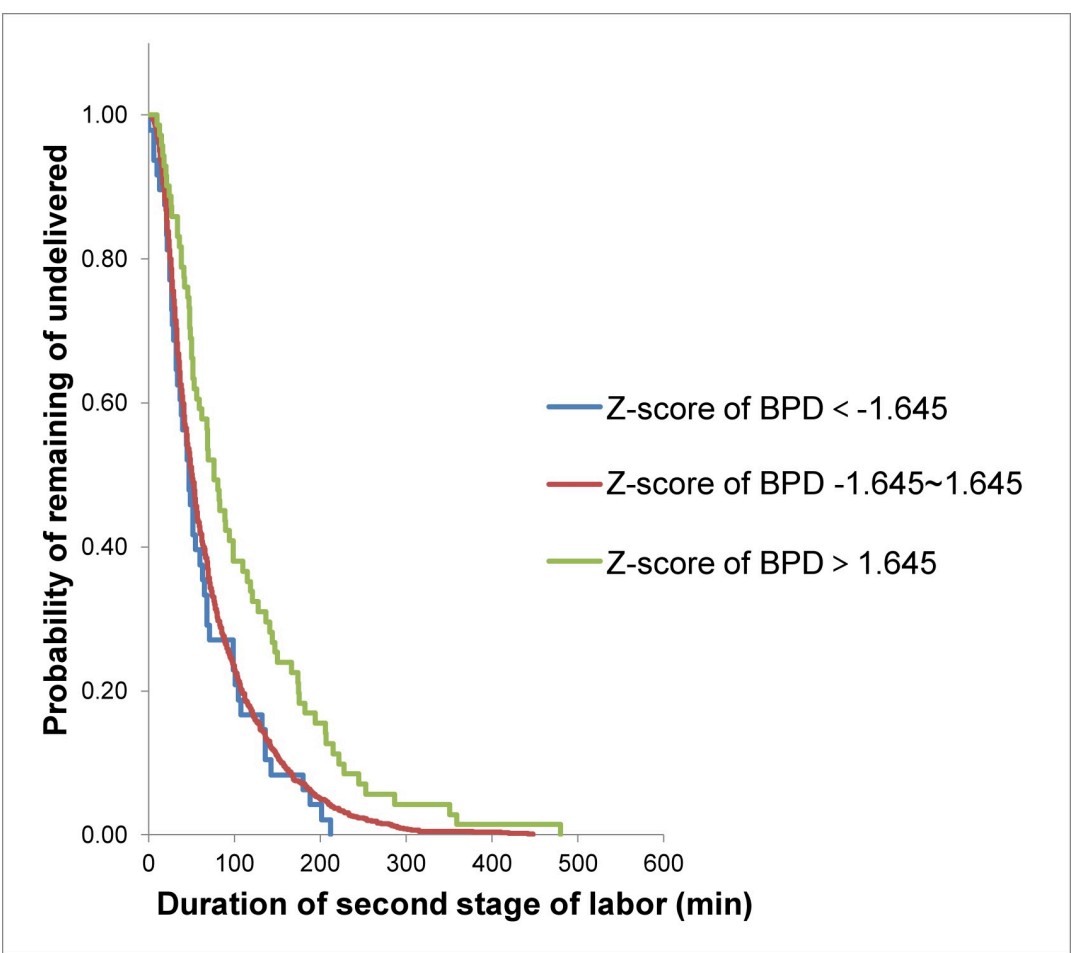

**Fig 1. Kaplan-Meier analysis of the duration of second stage of labor according to Z-score of BPD in nulliparous women.**

In multivariable analyses, the aOR for prolonged second stage of labor was 1.17 (95% confidence interval [CI], 0.89 to 1.53; P = 0.26) for each 1 increase in Z-score of BPD. In brief, the Z-score of BPD was not significantly associated with prolonged second stage of labor. On the contrary, macrosomia (aOR, 2.57; 95% CI, 1.46–4.56) was found to be associated with prolonged second stage of labor (Table 8).

Using the Kaplan-Meier survival analysis (Fig 2), at each time point during the second stage of labor, the percentage of women who had not yet delivered was not different between those who delivered neonates with large Z-scores of BPD and those who delivered neonates with smaller Z-scores of BPD (median duration of second stage of labor 13.1 min vs 18.9 min vs 17.7 min, P = 0.15).

## Discussion

In this retrospective study on Japanese women, we found that there is a significant effect of BPD on second stage of labor in only nulliparity. In short, a large BPD was significantly associated with prolonged second stage of labor after adjusting for potential confounding factors, and the Z-score of BPD was inversely correlated with the duration of the second stage of labor in only nulliparity.

**Table 6. Baseline characteristics of multiparous women in the study population.**

| | Normal second stage of labor (n = 1307) | Prolonged second stage of labor (n = 63) | P-value |
|---|---|---|---|
| **Maternal age, years** | 32.3 ± 4.8 | 34.4 ± 3.9 | 0.002 |
| **Gestational age, weeks** | 39.2 ± 1.03 | 39.6 ± 0.90 | 0.003 |
| **Maternal height, cm** | 158.5 ± 5.4 | 158.5 ± 5.4 | 0.94 |
| **Pre-pregnancy BMI, kg/m$^2$** | 21.2 ± 3.5 | 20.9 ± 2.8 | 0.81 |
| **IVF** | 31 (2.4) | 5 (7.9) | 0.02 |
| **Birth weight, g** | 3112 ± 370.7 | 3306 ± 379.9 | < 0.001 |
| **HDP** | 32 (2.4) | 2 (3.2) | 0.66 |
| **GDM** | 88 (6.7) | 3 (3.2) | 0.43 |
| **Instrumental delivery** | 5 (0.04) | 2 (3.2) | < 0.001 |
| **Pre-treatment with a uterine contraction agent** | 253 (19.4) | 26 (41.2) | < 0.001 |
| **Male infant** | 620 (47.4) | 38 (60.3) | 0.045 |
| **Duration of second stage of labor, min** | 14.6 ± 11.0 | 103.9 ± 73.5 | < 0.001 |
| **Fourth-degree perineal lacerations** | 0 (0.0) | 0 (0.0) | - |
| **Apgar score, 5 min** | 9.37 ± 0.61 | 9.33 ± 0.65 | 0.43 |
| **UA pH** | 7.33 ± 0.07 | 7.30 ± 0.06 | < 0.001 |

Values are presented as average ± standard deviation or as numbers (%).

BMI, body mass index; IVF, in vitro fertilization; HDP, hypertensive disorder of pregnancy; GDM, gestational diabetes mellitus; UA, umbilical arterial.

It seems natural to speculate that a large BPD, which may reflect a large fetal head, influences the progression of all stages of labor. However, in multiparity, Z-score of BPD was not associated with prolonged second stage of labor in this study. On the contrary, Salman L. et al showed that the larger the HC, which is an antepartum sonographic fetal head biometry parameter as well as BPD, the longer the second stage duration regardless of parity. The difference between these two research results may be attributed to two factors. First, the influence of molding, which is a physiological adaptation of the fetal head during its passage through the maternal pelvis for protection against fetal or maternal injury, is considered [14,21]. Bardin et al. reported that unlike the fixed fetal HC, the BPD is flexible to a certain degree owing to molding [14]. Second, in the clinical setting, it is obvious that there is a difference in ease of extension of the vagina and perineum between nulliparity and multiparity. Although further research is needed, these two factors can affect the relationship between the Z-score of BPD and duration of second stage of labor by parity.

Only a few studies to date have focused on the association between BPD and second stage of labor. Bardin et al. reported the relationship between BPD values and prolonged second stage of labor. The finding of Bardin et al.'s study is consistent with that of our study [14]. However, that study differed from our analysis as they did not separate nulliparity and multiparity and did not examine the association between BPD and second stage of labor in terms of

**Table 7. Prevalence of prolonged second stage of labor according to Z-score of fetal biparietal diameter quartiles measured within 7 days before delivery in multiparous women.**

| | Group 1 (Z-score of BPD; -5.63~-0.75) n = 343 | Group 2 (Z-score of BPD; -0.76~0.02) n = 331 | Group 3 (Z-score of BPD; 0.03~0.60) n = 355 | Group 4 (Z-score of BPD; 0.61~2.83) n = 341 | P-value |
|---|---|---|---|---|---|
| **Prolonged second stage of labor** | 9 (2.6) | 15 (4.5) | 17 (4.8) | 22 (6.5) | 0.12 |

Values are presented as numbers (%).

BPD, biparietal diameter.

**Table 8. Factors associated with prolonged second stage of labor in multiparous women.**

| Variables | Prolonged second stage of labor (n = 63) | Normal second stage of labor (n = 1,307) | Crude | | Adjusted | |
|---|---|---|---|---|---|---|
| | | | OR | 95% CI | OR | 95% CI |
| Z-score of BPD (median [25th–75th percentile]) | 0.31(-0.48–0.88) | 0.02 (-0.80–0.60) | | | 1.17 | 0.89–1.53 |
| Induction of labor | | | | | | |
| No | 56 | 1171 | 1.0 | Reference | 1.0 | Reference |
| Yes | 7 | 136 | 1.07 | 0.49–2.41 | 0.95 | 0.42–2.15 |
| Excessive maternal weight gain | | | | | | |
| No | 37 | 913 | 1.0 | Reference | 1.0 | Reference |
| Yes | 26 | 394 | 1.63 | 0.97–2.73 | 1.39 | 0.82–2.36 |
| Macrosomia | | | | | | |
| No | 41 | 1107 | 1.0 | Reference | 1.0 | Reference |
| Yes | 22 | 200 | 2.97 | 1.73–5.09 | 2.57 | 1.46–4.55 |

BPD, biparietal diameter.

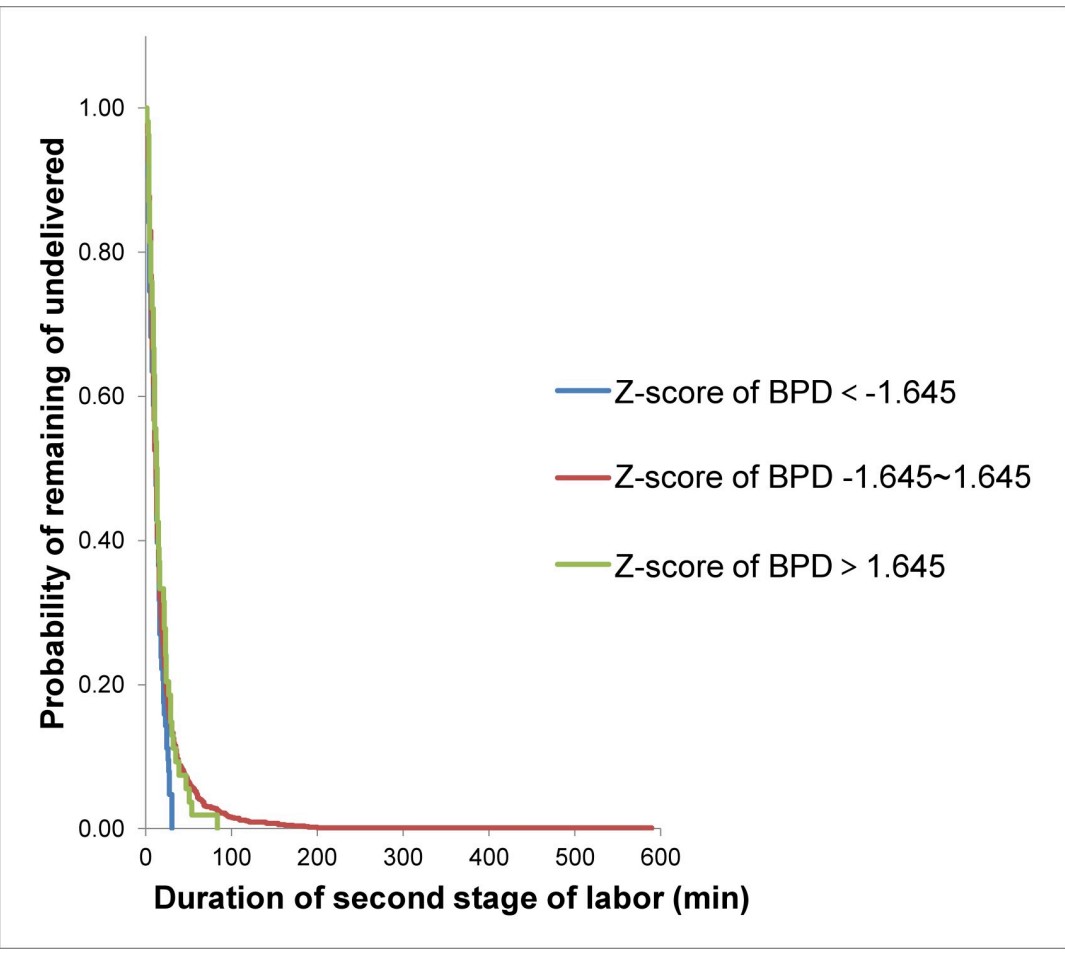

**Fig 2. Kaplan-Meier analysis of the duration of second stage of labor according to Z-score of BPD in multiparous women.**

the potential confounding factors for prolonged second stage of labor [14]. Compared to the previous study, the current study analyzed more risk factors for prolonged second stage of labor in a relatively large sample of Japanese women.

Macrosomia was also significantly associated with prolonged second stage of labor regardless of parity. To date, several studies have reported that macrosomia is one of the most important risk factors for prolonged second stage of labor. Therefore, in cases of suspected macrosomia, more attention should be paid to especially nulliparous women with large BPD during perinatal management. In this study, macrosomia with large BPD (Z-score of BPD > 1.645) caused prolonged second stage of labor in 9 of 19 (47.4%) cases in nulliparous women.

This study had certain limitations. Firstly, this was a single-center study, which may reduce the generalizability of our results to the general population. A large-scale multi-center, prospective cohort study is required to confirm our results in the general population. Secondly, the duration of second stage of labor was automatically calculated from the time when full dilatation was first documented until delivery. Therefore, this could be biased since full dilatation could have occurred earlier than when actually examined and documented. Third, we only included vaginal deliveries, and women who underwent cesarean section due to prolonged second stage of labor were not included in this study. This is because the time from full cervical dilation to the emergency cesarean section was not fully recorded in the medical records. Excluding those women may have led to a low estimation of the relationship between Z-score of BPD and prolonged second stage of labor, especially when performing survival analysis. Fourth, data regarding fetal occiput in a posterior or transverse position were not considered in this study [13], although this is a potential risk factor for prolonged second stage of labor.

## Conclusion

In conclusion, the Z-score of BPD was significantly associated with the prolonged second stage of labor in nulliparous women. Accurate risk stratification for prolonged second stage of labor using the Z-score of BPD could assist in the management of nulliparous women who are at risk of developing a prolonged second stage of labor.

## Acknowledgments

We would like to thank the study subjects for allowing the use of their personal data. We would also like to thank Editage (www.editage.com) for English language editing.

## Author Contributions

**Conceptualization:** Satoshi Shinohara.

**Data curation:** Satoshi Shinohara, Motoi Takizawa.

**Formal analysis:** Satoshi Shinohara, Kohta Suzuki.

**Investigation:** Satoshi Shinohara.

**Methodology:** Satoshi Shinohara.

**Project administration:** Satoshi Shinohara.

**Resources:** Satoshi Shinohara.

**Software:** Satoshi Shinohara.

**Supervision:** Satoshi Shinohara, Atsuhito Amemiya, Motoi Takizawa, Kohta Suzuki.

**Validation:** Satoshi Shinohara.

**Visualization:** Satoshi Shinohara.

**Writing – original draft:** Satoshi Shinohara.

**Writing – review & editing:** Satoshi Shinohara, Atsuhito Amemiya, Motoi Takizawa, Kohta Suzuki.

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
