## [Decision Letter · Decision Letter 0]

15 Jun 2020

PONE-D-20-13066

Fetal Biparietal Diameter as a Potential Risk Factor for Prolonged Second Stage of Labor: A Retrospective Observational Cohort Study

PLOS ONE

Dear Dr. Shinohara,

Thank you for submitting your manuscript to PLOS ONE. After careful consideration, we feel that it has merit but does not fully meet PLOS ONE’s publication criteria as it currently stands. Therefore, we invite you to submit a revised version of the manuscript that addresses the points raised during the review process.

SPECIFIC ACADEMIC EDITOR COMMENTS: Two expert reviewers in the field handled your manuscript. We thank them for their time. Although they found some interest in your study, there were numerous major concerns that arose during review. These comments include, but are not limited to: the need for clarity of various statements and definitions; questions about the statistical analysis; requirements to detail the number of patients included in the groups and the addition of subgroup analyses; the results need to be presented in a concise fashion; and the data need to support the conclusions for this to adhere to PLoS ONE's requirement of a scientific method-based manuscript. Please address ALL comments in your revised manuscript.

We look forward to receiving your revised manuscript.

Kind regards,

Frank T. Spradley

Academic Editor

PLOS ONE

The funders had no role in study design, data collection and analysis, decision to

publish, or preparation of the manuscript

Reviewers' comments:

Reviewer's Responses to Questions

**Comments to the Author**

1. Is the manuscript technically sound, and do the data support the conclusions?

Reviewer #1: Yes

Reviewer #2: No

2. Has the statistical analysis been performed appropriately and rigorously? 

Reviewer #1: I Don't Know

Reviewer #2: No

3. Have the authors made all data underlying the findings in their manuscript fully available?

Reviewer #1: Yes

Reviewer #2: Yes

4. Is the manuscript presented in an intelligible fashion and written in standard English?

Reviewer #1: Yes

Reviewer #2: Yes

5. Review Comments to the Author

Reviewer #1: Dear editor,

Many thanks for giving me this opportunity to review this attractive article. I look forward for further contribution to your imminent journal. Kindly find below the comments raised by my review.

1- In the abstract "abnormal labor progression" would be replaced by abnormality of labor progression.

2- In the abstract "We analyzed the incidence of prolonged second stage of labor and the association between biparietal diameter measured <7 days before delivery and prolonged second stage of labor" would be replaced by we analyzed the incidence of prolonged second stage of labor and its association with fetal biparietal diameter measured <7 days before delivery.

3- In the introduction, line 47, operator assists the delivery can be removed.

4- In the introduction, line 50, what do you mean by " after <37 weeks"? it is either after 37 weeks or before 37 weeks.

5- In the methods, lines 91-96, no need to mention indications for elective or emergency CS.

6- in the results, no data about fetal head position in women with prolonged second stage of labor. This would be considered as a major limitation.

7- Subgroup analysis for nulliparous and multiparous women would be recommended, in table 1 and 2.

8- The duration of the second stage of labor would be examined by survival methods. Any intervention as CS or instrumental delivery would affect the duration of labor.

9- Data about number of patients who required instrumental delivery or CS for arrest of the second stage of labor should be provided.

10- Data about episiotomy need and perineal tears would be provided.

11- Neonatal outcomes for both groups if possible.

Reviewer #2: Comments and questions

General comments

1. This is study is aimed to assess Fetal Parietal Diameter as a Potential Risk Factor for Prolonged Second Stage of Labor: A Retrospective Observational Cohort Study. Thank you for the opportunity to review this manuscript. I found the paper interesting; prolonged labor is an important issue which contributes majority of maternal death. I appreciate the authors really. This is Novel idea.

2. The manuscript is not described a technically sound piece of scientific research with data that supports the conclusions. For instance, the authors are talking about incidence. However, they fitted logistic regression to identify predictor variables which is totally wrong. The conclusions are not drawn appropriately based on the data presented and the statistical analysis has been performed appropriately and rigorously. Although the research question can be of interest in order to optimize tailored intervention on the problem, I do have several concerns that I will elaborate below.so, the language in submitted articles must be clear, correct, and unambiguous. Any typographical or grammatical errors should be corrected at revision, so please correct any specific error

Title

“Fetal Biparietal Diameter as a Potential Risk Factor for Prolonged Second Stage of Labor: A Retrospective Observational Cohort Study”. Here the authors have considered only one variable as risk factor for Prolonged Second Stage of Labor. Here I wander how the authors absolutely control other factors which seems impossible in observational data. From the best of my understanding, we can’t ever point out only one variable for an outcome from observational data. Rather we have to fit other variable in to the model and the result of our model will lead us to talk about risk factors. Therefore, I recommend the authors to correct the title as “Risk Factor for Prolonged Second Stage of Labor: A Retrospective Observational Cohort Study”. Even from the result null parity is one of the factor that associate with prolonged labor which is actually excluded first by your title.

Abstract:

1.Your abstract lacks some important information such as;

• some important result like incidence is not clearly stated. Does the author calculate incidence density or cumulative incidence? It would be make sense if they have well sated it as “incidence density or cumulative incidence”.

• Data collection tool, sampling procedure

• Model that you utilized

• Ethics

• conclusion

• Recommendation

• There are also grammatical and editorial issues

Introduction

1.general comment

Your background seems like good. But you did not state the problem well.

it lacks information such as;

• what is known about the problem in japan

• What is unknown about the problem?

2. Methods part

1. The subtitle “Study design” is not appropriate for the content that written under it. on the content there is study area, population

2. Based on your exclusion criteria the author excluded charts with missing data. why the authors did not manage it statically. Because charts that were excluded might have important variables like the outcome variable. This indirectly affect the incidence of prolonged labor. Therefore, I highly recommend you to use “multiple imputation” for analysis after you include these incomplete charts.

3. The second concern is why the authors excluded “Women with

multiple pregnancies”, and women went to emergency cesarean section? Once the women diagnosis for prolonged labor, they may go to emergency cesarean section. But they already diagnosed for prolonged labor. Therefore, excluding those women may lead to low estimation of the problem.

4. How Mann-Whitney U and Chi-squared tests could identify the potential confounding factors? I can be clear out with this.

5. The model; logistic regression model is not appropriate for the data. Please re analysis it using cox proportional hazard model.

Result

• Lacks brevity

• Mismatch between the result on abstract and on main body of the article. For example I cannot get texts that states about incidence on the main body of the paper.

• The author should summarize the result based on their objective.

• As to me the authors did not understand the nature of their data

• The study design is retrospective type of data but the method of analysis is for cross sectional type of data

• I can get whether the author want to know the incidence or the burden?

• Their objective is one “to evaluate the association between fetal 23

Bi parietal diameter and prolonged second stage of labor in a Japanese cohort”. But there are a lot of texts that are out of this objective.

• Finally, I strongly recommend authors to consult statscian to choose appropriate method of analysis for their data.

Discussion

Your discussion totally misses the implication of your finding.

6. PLOS authors have the option to publish the peer review history of their article (what does this mean?). If published, this will include your full peer review and any attached files.

Reviewer #1: No

Reviewer #2: Yes: Getenet Dessie

---

## [Author Response · Author response to Decision Letter 0]

20 Jul 2020

July 17, 2020

Zhong-Cheng Luo

Academic Editor

PLOS ONE

Dear Editor:

I, along with my co-authors, would like to re-submit the attached manuscript, titled “Fetal Biparietal Diameter as a Potential Risk Factor for Prolonged Second Stage of Labor: A Retrospective Observational Cohort Study” as an original research article (manuscript ID: PONE-D- 20-13066).

We appreciate your valuable comments, which have been very useful to us in improving the quality of our manuscript. All changes are shown in red in the revised manuscript. The entire manuscript has also been rechecked, and the necessary changes have been made in accordance with your suggestions. Our point-by-point responses to all of your comments have been prepared and provided below. Moreover, according to reviewer’s comment, we consulted with Professor Kohta Suzuki, who is a specialist in epidemiology, biostatistics, and clinical research. He belongs to the Department of Health and Psychosocial Medicine, Aichi Medical University School of Medicine.

We believe that the changes made based on your comments have significantly improved our manuscript, and we hope that you will find it suitable for publication in PLOS ONE.

Sincerely,

Dr. Satoshi Shinohara

Department of Obstetrics and Gynecology

National Hospital Organization Kofu National Hospital

11-35 Tenjin, Kofu, Yamanashi 400-8533, Japan

Tel: +81-55-253-6131

Fax: +81-55-251-5597

Email: shinohara617@gmail.com

 

RESPONSE TO REVIEWER 1

Comment 1

In the abstract "abnormal labor progression" would be replaced by abnormality of labor progression

Response 1

Thank you for your comment. According to your comment, we changed from "abnormal labor progression" to” abnormality of labor progression” in the abstract and introduction section. 

Comment 2

In the abstract "We analyzed the incidence of prolonged second stage of labor and the association between biparietal diameter measured <7 days before delivery and prolonged second stage of labor" would be replaced by we analyzed the incidence of prolonged second stage of labor and its association with fetal biparietal diameter measured <7 days before delivery.

Response 2

Thank you for your comment. According to your comment from "We analyzed the incidence of prolonged second stage of labor and the association between biparietal diameter measured <7 days before delivery and prolonged second stage of labor" to “we analyzed the incidence of prolonged second stage of labor and its association with Z-score of fetal biparietal diameter measured <7 days before delivery by parity.” 

Comment 3

In the introduction, line 47, operator assists the delivery can be removed.

Response 3

Thank you for your comment. According to your comment, we removed “operator assists the delivery” from the introduction. 

Comment 4

In the introduction, line 50, what do you mean by " after <37 weeks"? it is either after 37 weeks or before 37 weeks.

Response 4

Thank you for your comment. We changed from “after <37 weeks” to “earlier than 37 weeks”.

Comment 5

In the methods, lines 91-96, no need to mention indications for elective or emergency CS

Response 5

Thank you for your comment. According to your comment, we removed indications for elective or emergency CS.

Comment 6

in the results, no data about fetal head position in women with prolonged second stage of labor. This would be considered as a major limitation.

Response 6

Thank you for your comment. We agree with your opinion. However, there is no detailed description of the fetal head position in the electronic medical record. Since this is a retrospective study, it is not possible to obtain information that was not described in the electronic medical record. There were several limitations to this research. Based on this result, we plan to conduct a prospective study with the same theme in the future. At that time, we will consider the information on the fetal head position. In this study, information about the fetal head position was described as “Fourth, data regarding fetal occiput in a posterior or transverse position were not considered in this study, although this is a potential risk factor for prolonged second stage of labor.” in the limitation section. 

Comment 7

Subgroup analysis for nulliparous and multiparous women would be recommended, in table 1 and 2.

Response 7

According to your comment, we performed a subgroup analysis for nulliparous and multiparous women. Please look at the Results section for details.

Comment 8

The duration of the second stage of labor would be examined by survival methods. Any intervention as CS or instrumental delivery would affect the duration of labor.

Response 8

Thank you for your comment. According to your comment, as secondary object of this study, we examined the duration of the second stage of labor of nulliparity and multiparity by Kaplan-Meier survival analysis and log-rank test. Moreover, we agree with your opinion that Any intervention as CS or instrumental delivery would affect the duration of labor. However, we could not include women who had a cesarean section due to labor arrest in this study. Therefore, we added, “Third, we only included vaginal deliveries, and women who underwent cesarean section due to prolonged second stage of labor were not included in this study. This is because the time from full cervical dilation to the emergency cesarean section was not fully recorded in the medical record. Excluding those women may lead to a low estimation of the relationship between z-score of BPD and prolonged second stage of labor, especially when performing survival analysis.” in the limitation sections. 

Comment 9

Data about number of patients who required instrumental delivery or CS for arrest of the second stage of labor should be provided.

Response 9

Thank you for your comment. Please see for CS due to the arrest of the second stage of labor in response 8. Moreover, we could not check-up the exact number of instrumental delivery due to the arrest of the second stage of labor from medical records.

Comment 10

Data about episiotomy need and perineal tears would be provided.

Response 10

Thank you for your comment. According to your comment, we added data about “Fourth-degree perineal lacerations” in Table 2. However, we could not find the exact numbers of episiotomy and third-degree perineal lacerations from the medical records.

Comment 11

Neonatal outcomes for both groups if possible

Response 11

Thank you for your comment. We added, “Ap score” and “UA pH” in Table 2.

RESPONSE TO REVIEWER 2

Comment 1

This is study is aimed to assess Fetal Parietal Diameter as a Potential Risk Factor for Prolonged Second Stage of Labor: A Retrospective Observational Cohort Study. Thank you for the opportunity to review this manuscript. I found the paper interesting; prolonged labor is an important issue which contributes majority of maternal death. I appreciate the authors really. This is Novel idea.

Response 1

Thank you for your positive comment.

Comment 2

The manuscript is not described a technically sound piece of scientific research with data that supports the conclusions. For instance, the authors are talking about incidence. However, they fitted logistic regression to identify predictor variables which is totally wrong. The conclusions are not drawn appropriately based on the data presented and the statistical analysis has been performed appropriately and rigorously. Although the research question can be of interest in order to optimize tailored intervention on the problem, I do have several concerns that I will elaborate below.so, the language in submitted articles must be clear, correct, and unambiguous. Any typographical or grammatical errors should be corrected at revision, so please correct any specific error.

Response 2

Thank you for your comment. According to Reviewer 1 and 2, we revised our manuscript. Moreover, according to your comment, when we modified our manuscript, we consulted with Professor Kohta Suzuki, who is a specialist in epidemiology, biostatistics, and clinical research. He belongs to the Department of Health and Psychosocial Medicine, Aichi Medical University School of Medicine.

Comment 3

Fetal Biparietal Diameter as a Potential Risk Factor for Prolonged Second Stage of Labor: A Retrospective Observational Cohort Study”. Here the authors have considered only one variable as risk factor for Prolonged Second Stage of Labor. Here I wander how the authors absolutely control other factors which seems impossible in observational data. From the best of my understanding, we can’t ever point out only one variable for an outcome from observational data. Rather we have to fit other variable in to the model and the result of our model will lead us to talk about risk factors. Therefore, I recommend the authors to correct the title as “Risk Factor for Prolonged Second Stage of Labor: A Retrospective Observational Cohort Study”. Even from the result null parity is one of the factor that associate with prolonged labor which is actually excluded first by your title.

Response 3

Thank you for your comment. However, we did not consider only one variable (z-score of BPD) as a risk factor for Prolonged Second Stage of Labor. Therefore, a multivariable logistic regression model was used to identify variables significantly associated with prolonged second stage of labor. There was a significant relationship between the Z-score of BPD and prolonged second stage of labor. For the above reasons, we decided to leave the title of this paper as“ Fetal Biparietal Diameter as a Potential Risk Factor for Prolonged Second Stage of Labor: A Retrospective Observational Cohort Study.”

Comment 4

some important result like incidence is not clearly stated. Does the author calculate incidence density or cumulative incidence? It would be make sense if they have well sated it as “incidence density or cumulative incidence”.

Response 4

Thank you for your comment. We did not calculate incidence density and cumulative incidence. We only listed the total number of prolonged second stage of labor cases.

Comment and Response 5

1, Data collection tool, sampling procedure

 According to your comment, we added “Z-score of BPD, which was measured by experienced obstetricians using a GE Voluson ultrasound system (GE, Healthcare Japan, Tokyo, Japan)” in the abstract.

2, Model that you utilized

 We have already described as a retrospective observational cohort study in the abstract.

3, Ethics

As for Ethics, there is no space to write it in the abstract, so it is described in detail in study design section.

4, conclusion and Recommendation

According to your comment, we rewrote the conclusion and recommendation in the abstract.

5, There are also grammatical and editorial issues

Thank you for your comment. Based on your comment, the entire manuscript has been re-edited for language, grammar, and style.

Comment 6

Your background seems like good. But you did not state the problem well.

it lacks information such as;

• what is known about the problem in japan

• What is unknown about the problem?

Response 6

Thank you for your comment. We added, “This is because, especially in Japan, there has been little study done concerning second stage of labor and the risk factors for prolonged second stage of labor and evidence about the guide for the second stage of labor is insufficient.” in the introduction section. 

Comment 7

The subtitle “Study design” is not Since National Hospital Organization appropriate for the content that written under it. on the content there is study area, population

Response 7

Thank you for your comment. We added, “Kofu National Hospital is one of the tertiary perinatal care centers in Yamanashi Prefecture, relatively high-risk patients were included in this study.” in the study design section. 

Comment 8

Based on your exclusion criteria the author excluded charts with missing data. why the authors did not manage it statically. Because charts that were excluded might have important variables like the outcome variable. This indirectly affect the incidence of prolonged labor. Therefore, I highly recommend you to use “multiple imputation” for analysis after you include these incomplete charts

Response 8

Thank you for your comment. We could not obtain pre-pregnancy weight information for all women excluded as “missing data”. We consulted Prof. Kohta Suzuki about the needs for multiple imputation. He commented that multiple imputation is just a simulation study and not necessarily required if the missing data does not significantly affect the analysis results. Therefore, we performed multivariable logistic regression analysis except for data of “Excessive maternal weight gain” in nulliparous and multiparous women as sensitivity analyses (Table A and B). According to this additional analysis, we understood “Excessive maternal weight gain” did not significantly affect the study results. For the above reasons, we did not perform any additional analysis with multiple imputation.

Table A. Factors associated with prolonged second stage of labor in nulliparous women

Variables Prolonged second stage of labor (n=246) Normal second stage of labor (n=1095) Adjusted

 OR 95% CI

Z-score of BPD (median [25th–75th percentile]) 0.17

 (-0.43~0.88) 0.04

(-0.70~0.65) 1.18 1.02-1.37

Induction of labor 

No 193 912 1.0 Reference

Yes 53 183 1.34 0.95-1.90

Macrosomia 

No 209 1003 1.0 Reference

Yes 37 92 1.66 1.09-2.54

Table B. Factors associated with prolonged second stage of labor in multiparous women

Variables Prolonged second stage of labor (n=63) Normal second stage of labor (n=1307) Adjusted

 OR 95% CI

Z-score of BPD (median [25th–75th percentile]) 0.31 

(-0.48-0.88) 0.02

 (-0.80-0.60) 1.17 0.90-1.53

Induction of labor 

No 56 1171 1.0 Reference

Yes 7 136 0.95 0.42-2.14

Macrosomia 

No 41 1107 1.0 Reference

Yes 22 200 2.72 1.55-4.78

Comment 9

The second concern is why the authors excluded “Women with multiple pregnancies”, and women went to emergency cesarean section? Once the women diagnosis for prolonged labor, they may go to emergency cesarean section. But they already diagnosed for prolonged labor. Therefore, excluding those women may lead to low estimation of the problem.

Response 9

Thank you for your comment. In our hospital, all women with multiple pregnancies were delivered by elective cesarean section. Moreover, we agree with your opinion that the emergency cesarean section would affect the duration of labor. However, we could not include women who had a cesarean section due to labor arrest in this study. Therefore, we added “Third, we only included vaginal deliveries, and women who underwent cesarean section due to prolonged second stage of labor were not included in this study. This is because the time from full cervical dilation to the emergency cesarean section was not fully recorded in the medical record. Excluding those women may lead to a low estimation of the relationship between z-score of BPD and prolonged second stage of labor, especially when performing survival analysis.” in the limitation sections. 

Comment 10

How Mann-Whitney U and Chi-squared tests could identify the potential confounding factors? I can be clear out with this.

Response 10

Thank you for your comment. According to your comment, we rewrote as follows; “the Mann-Whitney U and Chi-squared tests were used to evaluate the effect of potential confounding factors for prolonged second stage of labor. 

Comment 11

The model; logistic regression model is not appropriate for the data. Please re analysis it using cox proportional hazard model.

Response 11

Thank you for your comment. According to your comment, we consulted Prof. Kohta Suzuki. He commented that Logistic analysis is the correct examination method in this study. This is because the purpose of this study is to investigate whether a clinically significant duration of second stage of labor is associated with BPD. Therefore, we did not analyze again using the cox proportional hazard model.

Comment 12

Result

• Lacks brevity

• Mismatch between the result on abstract and on main body of the article. For example I cannot get texts that states about incidence on the main body of the paper.

• The author should summarize the result based on their objective.

• As to me the authors did not understand the nature of their data

• The study design is retrospective type of data but the method of analysis is for cross sectional type of data

• I can get whether the author want to know the incidence or the burden?

• Their objective is one “to evaluate the association between fetal 23

Bi parietal diameter and prolonged second stage of labor in a Japanese cohort”. But there are a lot of texts that are out of this objective.

• Finally, I strongly recommend authors to consult statscian to choose appropriate method of analysis for their data.

Response 12

Thank you for your comment. According to your comment, we rewrote and summarized the results section to be concise. The incidence of second stage of labor was added in the result section separately described for nulliparity and multiparity. Moreover, according to your comment, we consulted Prof. Kohta Suzuki about the appropriate method of analysis for their data. 

Comment 13

Discussion

Your discussion totally misses the implication of your finding.

Response 13

Thank you for your comment. According to your comment, we rewrote the discussion section.

---

## [Decision Letter · Decision Letter 1]

27 Jul 2020

PONE-D-20-13066R1

Fetal Biparietal Diameter as a Potential Risk Factor for Prolonged Second Stage of Labor: A Retrospective Observational Cohort Study

PLOS ONE

Dear Dr. Shinohara,

Thank you for submitting your manuscript to PLOS ONE. After careful consideration, we feel that it has merit but does not fully meet PLOS ONE’s publication criteria as it currently stands. Therefore, we invite you to submit a revised version of the manuscript that addresses the points raised during the review process.

SPECIFIC ACADEMIC EDITOR COMMENTS: The reviewers still have some concerns about the definition of "incidence" as well as the model selection and the data analyses. Please address all concerns in your revised manuscript.

We look forward to receiving your revised manuscript.

Kind regards,

Frank T. Spradley

Academic Editor

PLOS ONE

Reviewers' comments:

Reviewer's Responses to Questions

**Comments to the Author**

1. If the authors have adequately addressed your comments raised in a previous round of review and you feel that this manuscript is now acceptable for publication, you may indicate that here to bypass the “Comments to the Author” section, enter your conflict of interest statement in the “Confidential to Editor” section, and submit your "Accept" recommendation.

Reviewer #1: (No Response)

Reviewer #2: All comments have been addressed

2. Is the manuscript technically sound, and do the data support the conclusions?

Reviewer #1: (No Response)

Reviewer #2: Yes

3. Has the statistical analysis been performed appropriately and rigorously? 

Reviewer #1: (No Response)

Reviewer #2: (No Response)

4. Have the authors made all data underlying the findings in their manuscript fully available?

Reviewer #1: (No Response)

Reviewer #2: No

5. Is the manuscript presented in an intelligible fashion and written in standard English?

Reviewer #1: (No Response)

Reviewer #2: Yes

6. Review Comments to the Author

Reviewer #1: (No Response)

Reviewer #2: incidence refers to the proportion or rate of persons who develop a condition during a particular time period. From the best of my understanding here in your data there is time to event data(starting from the onset of labour-----Time of delivery ) "Prolonged labour" by it self deals about time. Therefore, please think twice about your data and model selection . Even Just after you compete logistic regression Odds ratio is not appropriate measuring association for cohort study rather risk ratio sounds good.

7. PLOS authors have the option to publish the peer review history of their article (what does this mean?). If published, this will include your full peer review and any attached files.

Reviewer #1: No

Reviewer #2: No

---

## [Author Response · Author response to Decision Letter 1]

10 Aug 2020

August 08, 2020

Frank T. Spradley

Academic Editor

PLOS ONE

Dear Editor:

I, along with my co-authors, would like to re-submit the attached manuscript, titled “Fetal biparietal diameter as a potential risk factor for prolonged second stage of labor: a retrospective observational cohort study” as an original research article (manuscript ID: PONE-D- 20-13066).

We thank you and the reviewers for your thoughtful suggestions and insights, which have been very useful to us in improving the quality of our manuscript. All changes are shown in red in the revised manuscript. Our point-by-point responses to all reviewer comments have been prepared and provided below.

We believe that the changes made based on reviewer comments have significantly improved our manuscript, and we hope that you will find it suitable for publication in PLOS ONE.

Thank you for your consideration. I look forward to hearing from you.

Sincerely,

Dr. Satoshi Shinohara

Department of Obstetrics and Gynecology

National Hospital Organization Kofu National Hospital

11-35 Tenjin, Kofu, Yamanashi 400-8533, Japan

Tel: +81-55-253-6131

Fax: +81-55-251-5597

Email: shinohara617@gmail.com

 

RESPONSE TO REVIEWER 2

Comment 1

incidence refers to the proportion or rate of persons who develop a condition during a particular time period. From the best of my understanding here in your data there is time to event data(starting from the onset of labour-----Time of delivery ) "Prolonged labour" by it self deals about time. Therefore, please think twice about your data and model selection . Even Just after you compete logistic regression Odds ratio is not appropriate measuring association for cohort study rather risk ratio sounds good.

Response 1

Thank you for your comment. We have carefully reconsidered the analysis method based on your suggestion. As stated in your comment, we think the Cox proportional hazard model is appropriate if we treat the second stage of labor as time to event data. However, in this study we focused on only the event of the prolonged second stage of labor. Similar to our study, there are several studies that have conducted logistic analyses focusing on only the event of the prolonged second stage of labor (1~2):

1. Lina Salman, Anat Shmueli, Amir Aviram, et al. The association between neonatal head circumference and second stage duration.: J Matern Fetal Neonatal Med. 2019 Dec;32(24):4086-4092.

2. Eyal Sheiner, Asnat Walfisch, Mordechai Hallak, et al. Length of the second stage of labor as a predictor of perineal outcome after vaginal delivery.: J Reprod Med. 2006 Feb;51(2):115-119.

Moreover, we rechecked the references (3-9 listed below) cited in our paper. According to these studies, there are only reports that have compared maternal and fetal morbidities between prolonged second stage labor groups and non-prolonged second stage of labor groups, and there is no study that has examined the relation between maternal and fetal morbidities with prolonged second stage of labor as a continuous variable. To date, there is insufficient evidence as to whether a longer the second stage of labor corresponds with a poor perinatal prognosis. Rather, it can be understood that if the second stage of labor is not delayed, it does not matter in the clinical setting. Therefore, we consider a logistic analysis to be the correct examination method in this study.

3. Le Ray C, Fraser W, Rozenberg P, Langer B, Subtil D, Goffinet F, et al. Duration of passive and active phases of the second stage of labour and risk of severe postpartum haemorrhage in low-risk nulliparous women. Eur J Obstet Gynecol Reprod Biol 2011;158: 167–172.

4. Stephansson O, Sandström A, Petersson G, Wikström AK, Cnattingius S. Prolonged second stage of labour, maternal infectious disease, urinary retention and other complications in the early postpartum period. BJOG 2016;123: 608–616.

5. Cheng YW, Hopkins LM, Laros RK Jr, Caughey AB. Duration of the second stage of labor in multiparous women: maternal and neonatal outcomes. Am J Obstet Gynecol 2007;196: 585.e1–6.

6. Allen VM, Baskett TF, O'Connell CM, McKeen D, Allen AC. Maternal and perinatal outcomes with increasing duration of the second stage of labor. Obstet Gynecol 2009;113: 1248–1258.

7. Quiñones JN, Gómez D, Hoffman MK, Anath CV, Smulian JC, Skupski DW, et al. Length of the second stage of labor and preterm delivery risk in the subsequent pregnancy. Am J Obstet Gynecol 2018;219: 467.e1–467.e8.

8. Laughon SK, Berghella V, Reddy UM, Sundaram R, Lu Z, Hoffman MK. Neonatal and maternal outcomes with prolonged second stage of labor. Obstet Gynecol 2014;124: 57–67.

9. Sandström A, Altman M, Cnattingius S, Johansson S, Ahlberg M, Stephansson O. Durations of second stage of labor and pushing, and adverse neonatal outcomes: a population-based cohort study. J Perinatol 2017;37: 236–242.

Comment 2

In the abstract, line 38- 42. This is hard to read. It needs clarification.

Response 2

Thank you for your comment. According to your comment, we rewrote the relevant part as, “Kaplan-Meier survival analysis showed that at each time point during the second stage of labor, the percentage of women who had not yet delivered was higher among those who delivered neonates with large BPD Z-scores than among those who delivered neonates with smaller BPD Z-scores. ”.

Comment 3

Line 42, in the contrary, in multiparous women…., there was no mention for nulliparous women at all.

Response 3

Thank you for your comment. However, our data regarding nulliparous women were mentioned in the Abstract as follows: “In nulliparous women, multivariable analysis indicated that BPD Z-score was significantly associated with prolonged second stage of labor (adjusted odds ratio, 1.18; 95% confidence interval, 1.02–1.37).”.

Comment 4

Line 134, To identify the potential indicators of prolonged second stage of labor… could be removed and start the sentence with prolonged second stage of labor was defined as……

Response 4

Thank you for your comment. According to your suggestion, we removed “To identify the potential indicators of prolonged second stage of labor,” and replaced it with, “Prolonged second stage of labor was defined as…”.

Comment 5

Table 2 would be subdivided into nulliparous and multiparous women so that tables 3 and 6 would be presented in one table (table 2).

Response 5 

Thank you for your comment. Table 2 includes the baseline characteristics of the study population. We considered this information to be important for understanding the whole study population and have not changed it. We have, instead, added fourth-degree perineal lacerations, Ap score (5 min), and UA pH to Table 3 and Table 6 as well as Table 2.

Comment 6

line 334, Nulliparity should be replaced by nulliparous.

Response 6

Thank you for your comment. According to your suggestion, we changed the wording from nulliparity to nulliparous (page 27, line 331).

---

## [Decision Letter · Decision Letter 2]

31 Aug 2020

Fetal biparietal diameter as a potential risk factor for prolonged second stage of labor: a retrospective observational cohort study

PONE-D-20-13066R2

Dear Dr. Shinohara,

We’re pleased to inform you that your manuscript has been judged scientifically suitable for publication and will be formally accepted for publication once it meets all outstanding technical requirements.

Kind regards,

Frank T. Spradley

Academic Editor

PLOS ONE

Reviewers' comments:

Reviewer's Responses to Questions

**Comments to the Author**

1. If the authors have adequately addressed your comments raised in a previous round of review and you feel that this manuscript is now acceptable for publication, you may indicate that here to bypass the “Comments to the Author” section, enter your conflict of interest statement in the “Confidential to Editor” section, and submit your "Accept" recommendation.

Reviewer #1: All comments have been addressed

Reviewer #2: All comments have been addressed

2. Is the manuscript technically sound, and do the data support the conclusions?

Reviewer #1: Yes

Reviewer #2: Yes

3. Has the statistical analysis been performed appropriately and rigorously? 

Reviewer #1: Yes

Reviewer #2: Yes

4. Have the authors made all data underlying the findings in their manuscript fully available?

Reviewer #1: (No Response)

Reviewer #2: Yes

5. Is the manuscript presented in an intelligible fashion and written in standard English?

Reviewer #1: Yes

Reviewer #2: Yes

6. Review Comments to the Author

Reviewer #1: (No Response)

Reviewer #2: My comments have been fully addressed . The manuscript is much improved and suitable for publication .

Thank you!

7. PLOS authors have the option to publish the peer review history of their article (what does this mean?). If published, this will include your full peer review and any attached files.

Reviewer #1: No

Reviewer #2: **Yes: **Getenet Dessie

---

## [Editor Report · Acceptance letter]

14 Sep 2020

PONE-D-20-13066R2 

Fetal biparietal diameter as a potential risk factor for prolonged second stage of labor: a retrospective observational cohort study 

Dear Dr. Shinohara:

I'm pleased to inform you that your manuscript has been deemed suitable for publication in PLOS ONE. Congratulations! Your manuscript is now with our production department. 

Kind regards, 

on behalf of

Dr. Frank T. Spradley 

Academic Editor

PLOS ONE